

# Dynamic Analysis of Flowlike Landslides at Brienz/Brinzauls, Graubünden, Switzerland

Jordan Aaron[1], Larissa de Palézieux[1], Jake Langham[2], Valentin Gischig[1], Reto Thoeny[3] and Daniel Figi[3]

[1] Geological Institute, ETH Zurich, Zurich, Switzerland

[2] Department of Mathematics and Manchester Centre for Nonlinear Dynamics, University of Manchester, Oxford Road, Manchester M13 9PL, UK

[3] BTG Büro für Technische Geologie AG, Sargans, Switzerland

*Correspondence to*: Jordan Aaron (Jordan.aaron@eaps.ethz.ch)

**Abstract.** Accurate forecasting of the risk posed by catastrophic failure of rock slopes requires estimates of the potential impact area and emplacement velocity. While most previous work in this context has focused on rock avalanche behaviour, recent and well documented case histories are showing that a more diverse range of landslide classes can occur. In the present work, we analyse two rock slope failures that occurred at Brienz/Brinzauls in Switzerland. These events initiated within 500 m of each other on the same slope, but emplaced with velocities that differed by five orders of magnitude. We describe the derivation and implementation of a GPU accelerated numerical model that can simulate emplacement velocities on the order of m/day. We then perform forensic back-analysis of the two case histories. Our results highlight the role of path material in controlling emplacement behaviour, as well as the effect of moderate changes in source material lithology. We argue that these cases can form the foundation of more accurate hazard and risk analyses at similar sites, where a wider range of potential future behaviour than is typical should be considered.

## 1 Introduction

When a moving rock slope threatens a village, decision makers are required to forecast how far and how fast potential failures may travel. The diversity of emplacement behaviour that can be exhibited by failed rock slopes makes this a challenging task. In particular, emplacement velocities can range from moderate to rapid (cm to m/day) (Ranalli et al., 2010) to extremely-rapid velocities (>5 m/s) (Hungr et al., 2014). This seven order of magnitude velocity variation must be accounted for when generating and analysing potential failure scenarios in the context of a risk analysis, however a lack of field evidence combined with numerical modelling challenges has limited our ability to accurately analyse such cases. This has been made salient by the ongoing landslide hazard investigation taking place at a large landslide complex upon which the Swiss village of Brienz/Brinzauls, Graubunden, is located (Häusler et al., 2022; Kenner et al., 2022, 2025; Loew et al., 2024). The slopes near the village display evidence of both extremely-rapid and moderate-to-rapid emplacement dynamics, and studying these events provides a unique opportunity to advance our understanding of rock slope emplacement behaviour, as is done herein.



Previous work on flowlike landslides in rock has primarily focused on understanding the motion of rock avalanches, which are extremely-rapid flows of fragmented rock initiated from a large rock slope failure (Aaron et al., 2020, p. 202; Aaron & McDougall, 2019; D. Cruden & Hungr, 1986; Dufresne et al., 2016; Hungr et al., 2014; Hungr & Evans, 2004). Many different numerical models that can simulate the motion of rock avalanches have been proposed in literature (Aaron & Hungr, 2016;

Hungr, 1995; Mangeney-Castelnau, 2003; McDougall & Hungr, 2004; Preuth et al., 2010). The most common of these approaches, termed the 'equivalent fluid approach' by Hungr (1995) treats the failed mass as a simple fluid, whose behaviour is governed by user selected rheologies, with parameters calibrated based on past case histories (Aaron & McDougall, 2019; Preuth et al., 2010; Sosio et al., 2012). The combination of field evidence and numerical modelling has been used to demonstrate that rock avalanches can move with dynamic friction angles much less than 30°, which is the value that would be

expected for dry granular flows of fragmented rock. This phenomenon has been termed 'excessive mobility', and is relevant to the two case histories analysed herein ( e.g. Aaron & McDougall, 2019; Davies et al., 1999; Dufresne et al., 2016; Heim, 1932; Hungr & Evans, 2004; Li, 1983, p. 198; Scheidegger, 1973; Whittall et al., 2017).

Explanations for this phenomenon have relied on both intrinsic and extrinsic factors (e.g. Hungr & Evans, 2004) and the cause for it is still debated. Of particular relevance to the present work are extrinsic theories which rely on the character of the source

zone and path material to explain mobility (e.g. Aaron et al., 2017, 2022; Hungr & Evans, 2004; Sosio et al., 2012). As summarized in Hungr & Evans (2004), these theories state that rock avalanches which interact with weak path material (such as liquefiable substrate or glacial ice) experience the highest mobility. This was further analysed to argue that mechanisms may also act in the source zone to increase mobility (Aaron et al., 2022; Aaron & McDougall, 2019). However, few researchers have attempted to investigate these phenomena in the context of the full complexity that can be exhibited by flowlike

landslides, which can include variations in source lithology.

In contrast to the wide body of literature available regarding rock avalanche motion, moderate to rapid flowlike movements (defined by velocities in the m/month to m/day range) in rock have received considerably less attention in the literature. Flowlike landslides with these velocities have mainly been studied in the context of earthflows, where an emphasis is placed on understanding surging behaviour over yearly and decadal timescales (e.g. Aaron et al., 2021; Mackey & Roering, 2011;

Nereson & Finnegan, 2018; Vassallo et al., 2016). Further, many slow creeping landslides in rock have been documented (e.g. Crosta et al., 2013; Ranalli et al., 2010; Wolter et al., 2020). The motion of these landslides has been simulated using simplified 1D numerical models, which include a viscous component in the basal resistance law (e.g. Ranalli et al., 2010; Rutter & Green, 2011). In the present work, large failures in clay-rich rocks that transition into flowlike landslides (termed 'slump-earthflow' in Varnes (1978)) are particularly relevant. These can have a morphology of rotational/translational slide deposits near the

source zone, which transition into earthflows at their distal end (Skempton et al., 1997). Although these have been described in literature, few models exist that can simulate their motion over complex terrain have been proposed. This is because viscous type drag laws typically require small timesteps to maintain numerical stability, which when combined with multi-year



emplacement times can lead to simulation times of decades to centuries. A lack of suitable numerical models, combined with limited field observations, has resulted in a poor understanding of the factors governing their occurrence.

Source lithology is one potential reason why some rock slope failures reach extremely-rapid velocities, whereas others only reach slow velocities (Ranalli et al., 2010; Wolter et al., 2020). It is known from the rock mechanics literature that a diverse range of shear behaviour can be observed, ranging from high strength, brittle response in hard rocks such as dolomite (Evans et al., 1994, p. 199; Poschinger et al., 2006), to low strength, ductile response in soft rocks (Skempton et al., 1997). It can be expected that this would lead to different emplacement dynamics for flowlike landslides, however a lack of direct field
observations has been missing that would enable the systematic study of these phenomenon.

In the present work we perform a detailed back-analysis of two well-documented landslides that have occurred at Brienz/Brinzauls within 500 m of each other. These landslides, named lgl Rutsch and Insel, were both large volume, flowlike landslides that share superficial similarities, but displayed distinct dynamics. In particular, despite similar runout distances and impact areas, these two events exhibited emplacement velocities that differed by five orders of magnitude. Currently
available numerical tools are inadequate for simulating lgl Rutsch with velocities on the order of m/day. Thus, we also describe the implementation and validation of a new numerical model which can be used to simulate landslides that move with moderate to rapid velocities (velocities on the order of m/day).

## 2    Site Description

The Brienz/Brinzauls landslide is a large landslide complex located in the Canton of Graubunden, Switzerland. The entire
active landslide extends from the Albula river at about 860 m a.s.l. to a topographic bench at about 1800 m a.s.l. Signs of deep-seated slope deformation (scarps, back-scarps, etc) reaches up to the crest of Piz Linard (2768 m a.s.l.) (Loew et al., 2024) (Figure 1). The landslide is located at the intersection of the Penninic and Austroalpine nappes (Loew et al., 2024). The geology of the landslide complex has been described in detail in BTG AG (2022). It consists of clay-rich Flysch and Allgäu units overlain by rauhwacke/dolomite of the Raibler formation, and the competent dolomites of the Vallatscha formation
(Figure 1).

The active part of the landslide consists of approximately 170 Mm$^3$ of material moving on a compound rupture surface (BTG AG, 2022; Loew et al., 2024). This large landslide complex hosts many secondary processes, including rockfall (Schneider et al., 2023), as well as the flowlike landslide types analysed herein. Landslide movements on the order of 10 cm/year were detected already in the 1930s. After the observation of fresh tension cracks, the slope has been intensively monitored since
2011. Since then the overall velocity of the landslide complex has increased dramatically and is currently significantly greater than 0.5 m/year in most parts of the landslide (Loew et al., 2024).



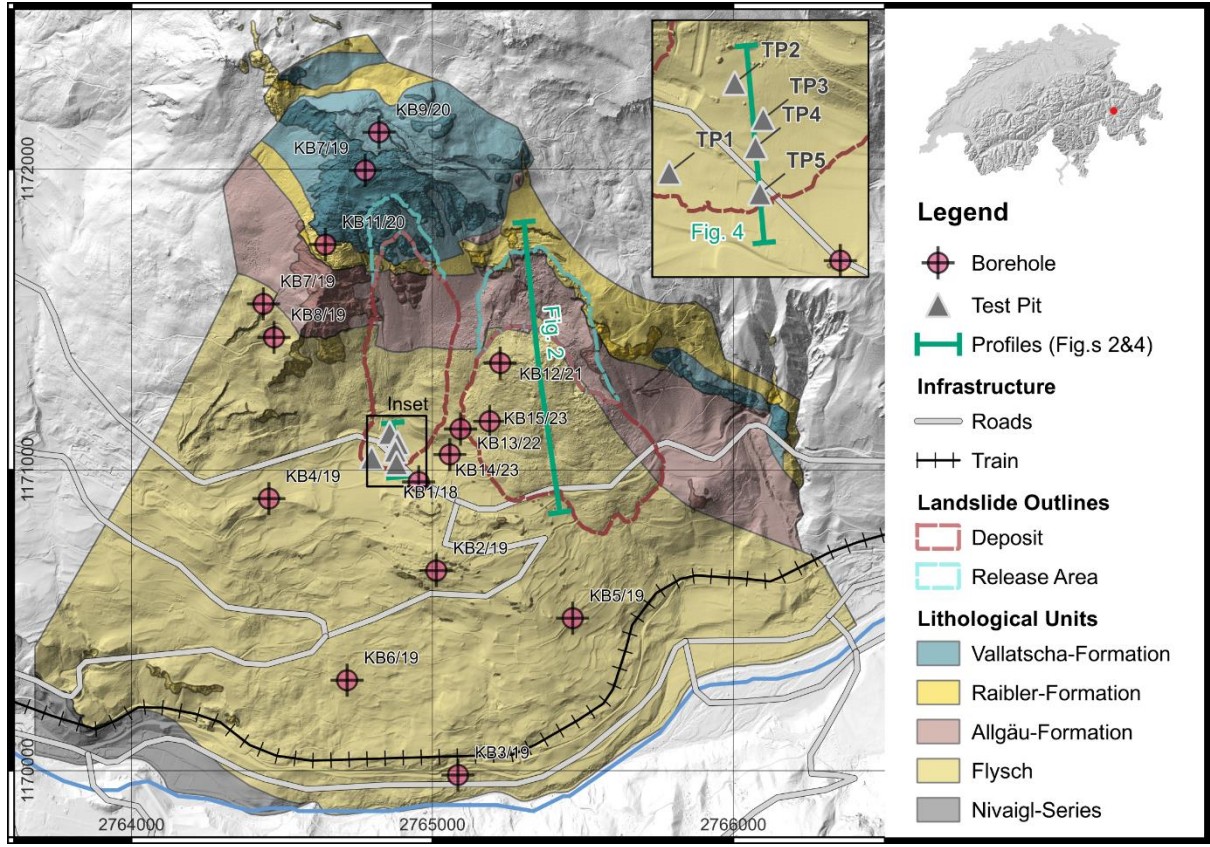

**Figure 1: Site overview, showing major geological units, the outlines of the two flowlike landslides, and borehole locations, (modified after BTG (2022); hillshade ©swisstopo).). The inset shows the section line used in Figure 4.**

## 2.1 lgl Rutsch Event Description and Topographic Reconstruction

Quantifying hazard and risk at Brienz/Brinzauls, as well as similar sites, requires an understanding of the full range of potential emplacement scenarios. Constraints on this can be obtained by looking at past landslide which have occurred at the site. The lgl Rutsch landslide (Figure 1) was a moderate to rapid flowlike landslide that occurred at Brienz/Brinzauls between 1879 and 1881 (Heim, 1881). Surface mapping and boreholes through this landslide (Figure 1) indicate that it initiated in the geological units of the Raibler and Allgäu formation, and transitioned from disturbed Allgäu schists in the proximal zone into clayey soil deposits at the distal toe, with thicknesses of 20.5 m and 30 m. We used this surface and subsurface data, as well as a geomorphic interpretation, to reconstruct the rupture surface and pre-failure source zone geometry of the lgl Rutsch landslide (Figure 2). In particular, a flat area, on which the village is founded, can be seen to run under the deposit (Figure 1). We adjusted the present-day contours to extrapolate this flat area under the deposit, and to maintain deposit thickness estimates consistent with the boreholes. We further interpreted the pre-failure geometry in the source zone, based on the present day



morphology of the head scarp. This led to a reconstructed deposit volume of 5.2 Mm$^3$, and source volume of 4.2 Mm$^3$, which results in a bulking factor of 1.2.

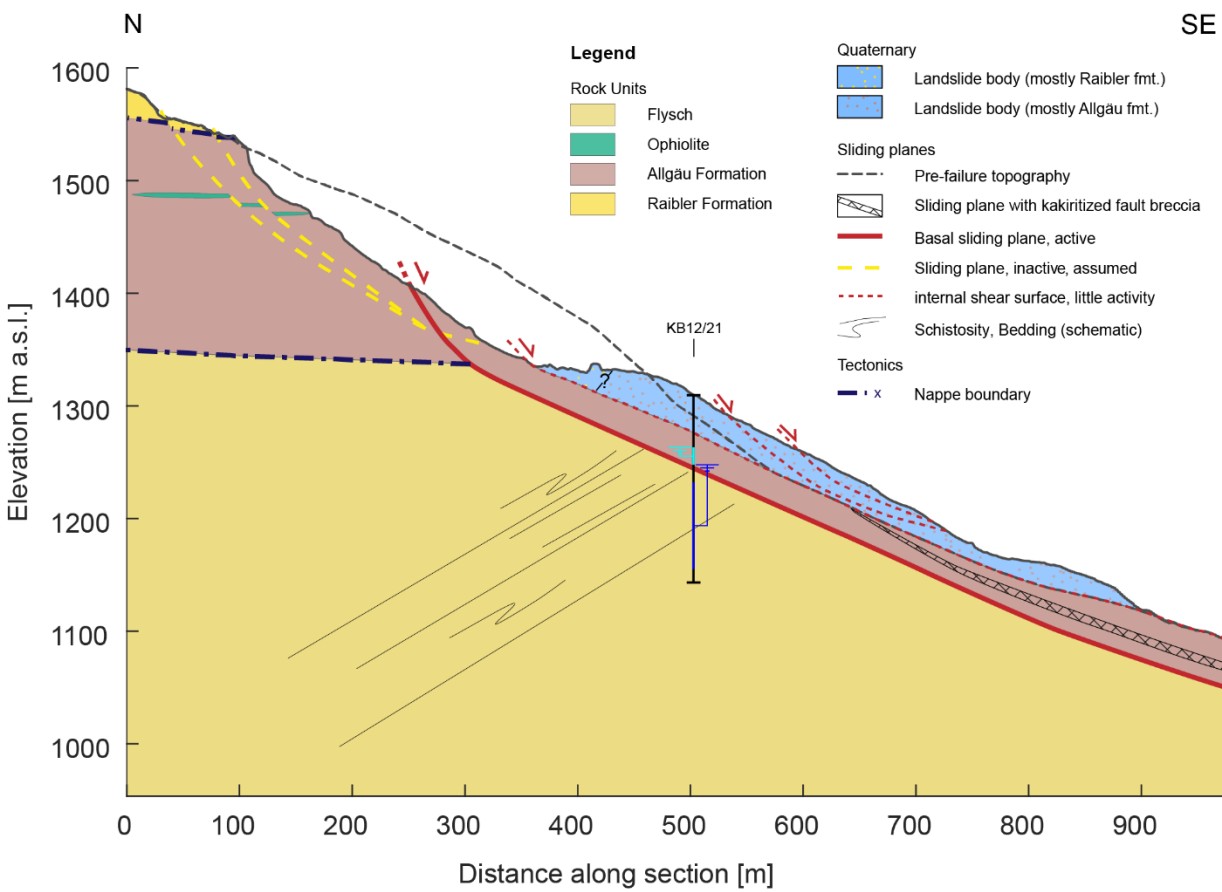

**Figure 2: Topographic reconstruction of the lgl Rutsch landslide (modified after BTG 2022)**

The emplacement of the lgl Rutsch landslide was documented by the Swiss geologist Albert Heim, in a report which describes the nearly three-year emplacement process (Heim, 1881). In his report, he compares the emplacement of the lgl Rutsch landslide to the movement of a glacier, both because of the internal deformation structures that developed during emplacement, as well as the slow, flowlike movement. Albert Heim's account includes some quantitative and qualitative indications of the landslide velocity, which we interpret and summarize on Figure 3. Heim (1881) describes the landslide as moving by 1 to 2

m/day in the winter of 1878. Following this, velocities are described more qualitatively, but clearly indicate that the landslide has slowed down on average, while continuing a long-term displacement trend with seasonal cycles (interpreted here as fluctuations between ~0.125 m/day and 0.75 m/day, albeit with significant uncertainty). Finally, Heim (1881) states that by the summer of 1881 the movement rate had become imperceptible. Heim (1881) suggested that the landslide would continue





to periodically move, potentially reaching the Albula river in a few hundred years. However, the landslide has remained

relatively stable for the nearly 150 years since movement ceased, with modern velocities on the order of a few cm/year.

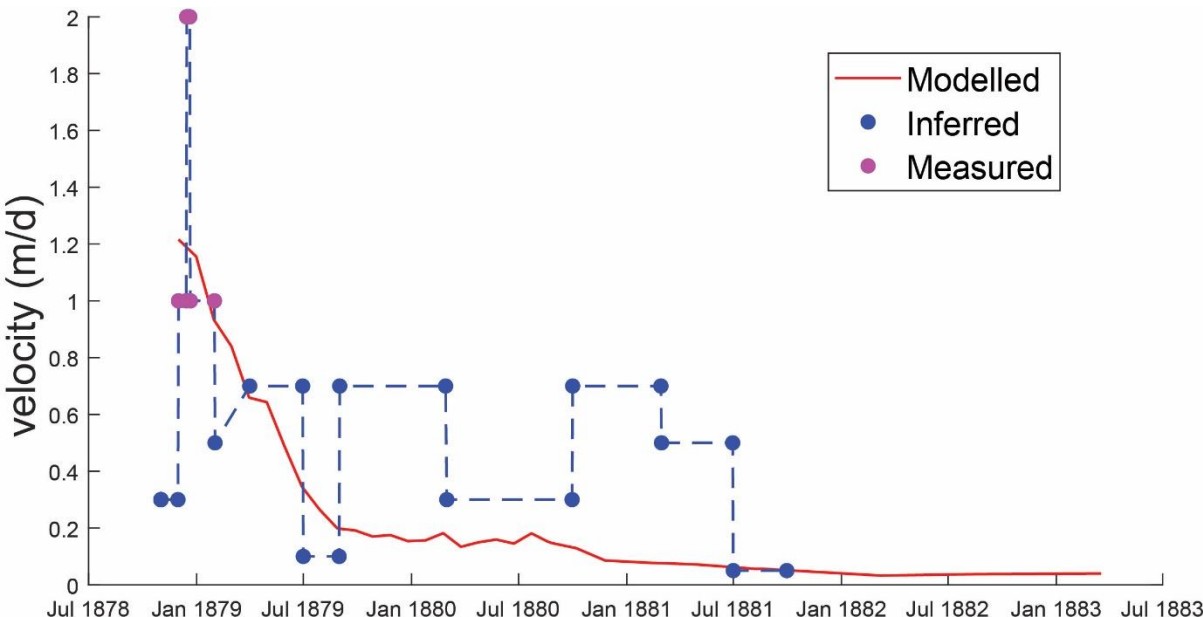

**Figure 3: Velocity data interpreted from qualitative reports of the lgl Rutsch landslide. The four magenta points indicate velocity estimates that are based on quantitative data presented by Heim (1881), whereas the blue points are based on qualitative indications.**

## 2.2  Insel Event Description and Topographic Reconstruction

The Insel failure, which occurred on June 15, 2023, strongly contrasts the slow, viscous behaviour exhibited by the lgl Rutsch. Starting in 2022, the Insel compartment became distinct from its surroundings, and started to accelerate towards the village of Brienz/Brinzauls. The installed early warning system indicated that the compartment was heading towards a catastrophic failure, so in mid May, 2023 the village was evacuated (Loew et al., 2024). The compartment then failed during a short period on the evening of June 15, 2023, and reached extremely-rapid velocities which were five orders of magnitude greater than lgl

Rutsch. Images of the deposit are shown on Figure 4A and B, and test pits through the toe of the deposit are primarily composed of colluvium derived from the Allgäu formation, with a thin mantling of blocks of Vallatscha (Figure 4C). The debris further upslope and in the source zone is primarily composed of colluvium derived from the Vallatscha formation (Loew et al., 2024).



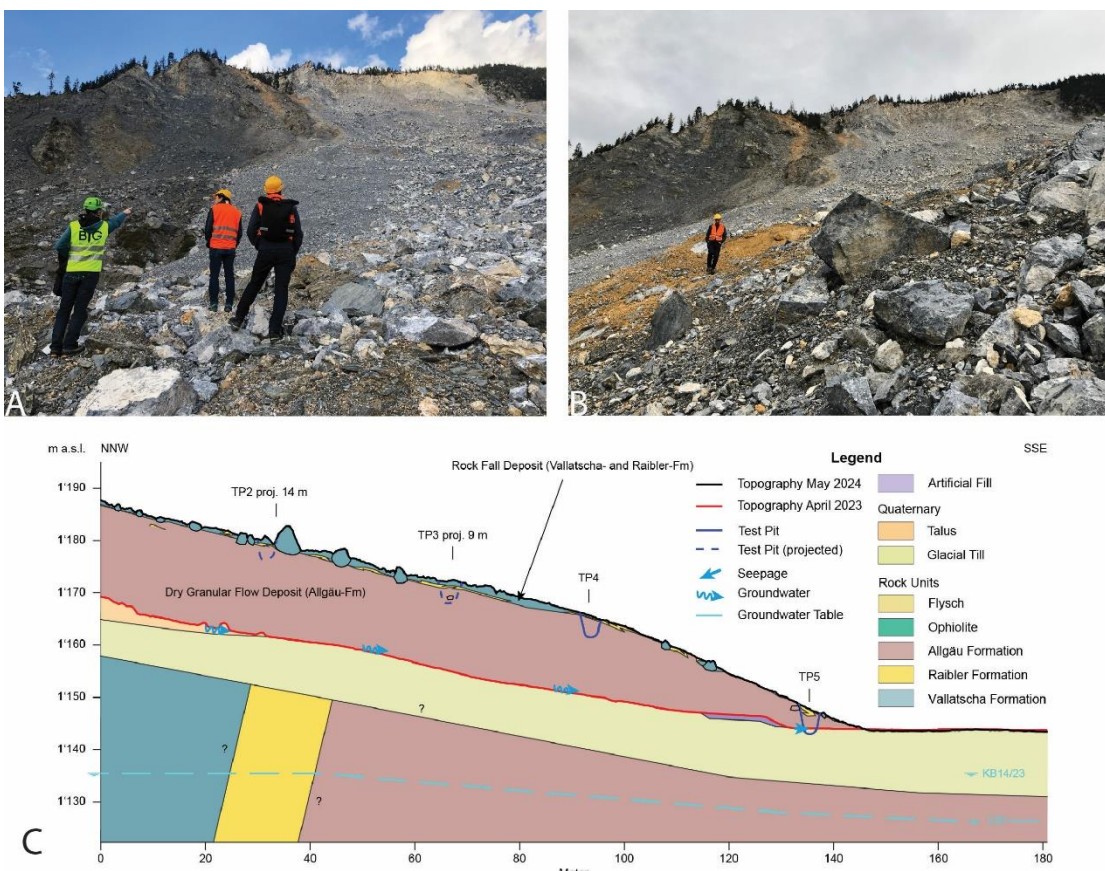

**Figure 4: Images of the deposit of the Insel event (A and B), as well as the interpreted stratigraphy of the deposit based on test pits (C). The locations of the section line used for C, as well as test pit locations, are shown on Figure 1.**

The Insel failure was documented by an extensive monitoring system, which included a georadar, seismic stations, and imagery from immediately before and after the main failure. This event occurred at night time, so no optical data of the failure exists. The seismic and radar data is plotted together on Figure 5. Our interpretation, supported later by numerical modelling, is that the failure mainly occurred during a 120 second period, albeit with many precursory events (i.e. small volume slides and rock fall at the toe of the failing Insel compartment). This can be seen by the strong increase in seismic normalized amplitudes, as well as the decorrelation pattern visible in the georadar data, in particular the downslope extension of the area that is decorrelated (Figure 5). The available monitoring data cannot distinguish the failure sequence within this 120 second period, and one main failure as well as two failures separated by a few tens of seconds are both viable hypotheses (tested with numerical modelling below).



**Figure 5: Seismic and radar data during the one hour period that contains the Insel failure. The color bar shows the phase of the radar data, it can be seen the spatial area that is decorrelated increases during the 2 minute period that we interpret as the main failure stage. The seismic data is from (Borgeat et al., 2025; Häusler et al., 2025; Swiss Seismological Service (SED) At ETH Zurich, 2012).**



High resolution topography of the site is available prior to and after the 2023 failure of the Insel compartment, however a significant volume of material is deposited on the rupture surface. For the dynamic models that follow, we used a 3D extrapolation of the interpreted rupture surface given by Loew et al. (2024). This results in a source volume of 2.1 Mm$^3$, and a deposit volume of 2.5 Mm$^3$.

## 3    Methods

In the present work we use the numerical model 'Orin-3D' as the computational framework with which to analyse the lgl Rutsch and Insel failures. Orin-3D is a GPU accelerated, depth-averaged Lagrangian model that solves the equations of motion using a parallel implementation of the smooth particle hydrodynamics (SPH) numerical method (Aaron, 2023). Orin-3D provides an over two orders of magnitude increase in computational efficiency, compared to a widely used equivalent fluid model (Aaron, 2023). The equations of motion solved by Orin-3D are

$$\rho h \frac{D v_x}{D t} = \rho h g_x - k_x \sigma_z \frac{\partial h}{\partial x} + \tau_{zx}, \tag{1}$$

$$\rho h \frac{D v_y}{D t} = \rho h g_y - k_y \sigma_z \frac{\partial h}{\partial y} \tag{2}$$

where $\frac{D}{Dt}$ represents the Lagrangian or material derivative, $\rho$ is the density of flowing material (assumed constant), $v_x$, $v_y$ are the depth-averaged $x$ and $y$ velocities, $h$ is the flow depth, $g_x$, $g_y$ are the x and y components of the gravity vector, $k_x$, $k_y$ are the x and y horizontal stress ratios (ratio of lateral stress to bed normal stress), $\sigma_z$ is the bed normal stress, $\tau_{zx}$ is the basal resistance and only is present in Eq [1] because the coordinate system is aligned with the local direction of motion. Note that only the final form of the equations is given here. More details are provided in the literature (Hungr & McDougall, 2009; McDougall & Hungr, 2004).

The basal resistance term ($\tau_{zx}$) is particularly relevant to the present work. This term is governed by a user specified rheology. The form and selected parameters exert a strong control on the velocity, flow depth and impact area of the simulated landslide. This term therefore accounts for all site-specific phenomena that occur during flowlike landslide emplacement, and calibrated parameters can be interpreted to reveal underlying mechanisms (e.g. Aaron et al., 2017; Aaron & McDougall, 2019). The most commonly used rheologies for flowlike landslides in rock are the Coulomb and Voellmy drag laws, given by

$$\tau_{zx} = -\sigma_z \tan(\varphi) \text{ and} \tag{3}$$



$$\tau_{zx} = -\left[\sigma_z \tan(\varphi) + \frac{\rho g v_x^2}{\xi}\right], \tag{4}$$

respectively, where $\varphi$ is the friction coefficient and $\xi$ is a constant dimensional turbulence coefficient, both of which are calibrated parameters.

The basal drag models in Eqs. (3 and (4 were developed to simulate extremely-rapid velocities, such as those exhibited by the Insel failure. They are thus inappropriate for simulating the moderate to rapid velocities documented for the lgl Rutsch landslide. We therefore implemented a new rheology that contains a viscous term, in order to allow the model to simulate velocities on the order of m/day. The rheology is based on Ranalli et al. (2010) that supplements Coulomb drag with a viscous term, in order to allow the model to simulate velocities on the order of m/day:

$$\tau_{zx} = -(\sigma_z \tan(\varphi) + \delta v_x), \tag{5}$$

where $\delta$ is a constant viscous constitutive parameter which is equal to the viscosity divided by the characteristic thickness of the shear zone, with units of Pa s/m. Following the original approach in Dan3D (McDougall & Hungr, 2004), Orin3D evaluates the bed-normal stress as:

$$\sigma_z = \rho h(g \cos(\alpha) + \frac{v_x^2}{R}), \tag{6}$$

where $\alpha$ and $R$ are (respectively) the angle and bed-normal radius of curvature of the local slope, measured along the direction of motion.

The inclusion of the second term in Eqn [5], combined with the three-year emplacement time of the lgl Rutsch event, renders Orin-3D's currently implemented explicit time stepping scheme too computationally expensive for simulating lgl Rutsch. This is because the time step restriction it imposes leads to projected simulations times on the order of decades to centuries. Consequently, we modified Orin-3D to treat the viscous component of the drag implicitly. This numerical strategy has been employed elsewhere to stabilize the numerical integration of shallow-layer models with stiff drag terms (e.g. Casulli, 1990; Chertock et al., 2015a, 2015b).

Specifically, we employ a semi-implicit Euler scheme, that updates velocities at the $(i + 1)$-th time step as follows

$$v_{x,i+1} = \left(1 - \Delta t \frac{\delta}{\rho h_i}\right)^{-1} \left[\left(\frac{\rho h_i g_{x,i} - k_{x,i}\sigma_{z,i}\frac{\partial h_i}{\partial x} - \sigma_{z,i}\tan(\varphi)}{\rho h_i}\right)\Delta t + v_{x,i}\right], \tag{7}$$

$$v_{y,i+1} = \left(1 - \Delta t \frac{\delta}{\rho h_i}\right)^{-1} \left[\left(\frac{\rho h_i g_{y,i} - k_{y,i}\sigma_{z,i}\frac{\partial h_i}{\partial y}}{\rho h_i}\right)\Delta t + v_{y,i}\right], \tag{8}$$

where $\Delta t$ is a constant time interval and quantities with subscripts depending on $i$ or $i + 1$ denote the evaluation of those fields at the corresponding step. These formulae discretize the stiff viscous component of the drag implicitly, while treating the



remaining terms explicitly. This method dramatically extends the size of stable time steps that may be taken, without introducing the complications presented by fully implicit schemes. Note that, because the direction of the flow can change relative to the Lagrangian frame at step $i$, both updates require the implicit drag treatment. Equations (7 and (8 are then spatially discretized and solved using the parallel implementation of smooth particle hydrodynamics that already exists in Orin3D.

### 3.1    Simulation Methodology

We back-analysed the lgl Rutsch and Insel case histories in order to derive model parameters that can be used for forecasting events that have similar types.  For lgl Rutsch, we used the new semi-implicit runout model (Equations (7 and (8), as well as the failure geometry shown on Figure 2. We used timesteps of 720 seconds, 4000 particles, a topographic resolution of 10 m, and performed a probabilistic calibration (described below) in order to calibrate the friction angle and viscous parameter (Equation [5]).  As part of this calibration, we compare measured and simulated velocities for the event (Figure 3).  The latter

velocities are obtained by calculating the velocity of the simulated front as it moves over the slope, as this provides the closest simulated quantity to the velocities estimated from eye witness accounts.

For the Insel event, we again calibrated using 4000 SPH particles and a topographic resolution of 5 m, but this time used the default explicit timestepping scheme in Orin3D.  For this event there is some uncertainty regarding the failure sequence, and it is possible that the main failure occurred as two events separated in time, or as one event.  In the present work, we simulated

two end-member scenarios:

1.  The event occurred in two separate stages, and there was no dynamic interaction between the two stages (ie the first stage deposits and is then overrun by the second stage).  Both stages had similar volumes.

2.  The majority of the source volume failed in one main stage.  This scenario doesn't exclude secondary failures, however their volume would have been much less than that of the main event.  This scenario is also representative

of a multi-stage scenario where the stages exhibit significant dynamic interaction (ie the first stage is still moving when the second stage initiates).

We then calibrated the model for both these scenarios using the Coulomb frictional and Voellmy rheologies (Equations (3 and (4).  Both rheologies were used in order to provide calibrated parameters that are comparable to past case histories.  The calibration results were assessed by comparing the simulated and observed deposit distributions and source zone lithology

deposit locations.  This latter observation was simulated by tracking the trajectories of individual particles in the SPH simulation.  We then analysed where particles, which initiate in the Valatscha and Allgäu deposit zones, are simulated to deposit, and compared this to field observations of the deposit.  We note that our numerical model is depth averaged, and each



particle represents a substantial volume of material. The simulated deposit distribution is therefore representative of average behaviour, and cannot resolve smaller scale details such as thin surface deposits (Figure 4).

## 3.2 Calibration Methodology

The main goal of the back-analyses performed in the present work is to derive basal resistance parameters that can be interpreted to infer flow mechanics. It is therefore important that uncertainties in the calibrated parameters are quantified, such that they can be considered in forecasts. To do this, we used the calibration methodology that is detailed in Aaron et al. (2019), which can be used to estimate a posterior probability density function of the unknown bulk material parameters that govern the user-specified rheologies. Briefly, this methodology uses estimates of impact area, deposit distribution and velocity to calculate a model fitness number for a given input parameter set. We then run the model a large number of times on a regular grid of parameters, in order to calculate the fitness numbers for a discretization of the entire parameter space, and use these numbers and assumed standard deviations to calculate a posterior probability density function. The parameter ranges explored for the various rheologies are given in Table 1.

**Table 1: Best fit basal resistance parameters for the simulated scenarios**

| Event | Friction Angle ($\varphi$) for Eq [3] and [5]. | Voellmy Friction Coefficient $(\tan(\varphi))$ for Eq [4]. | Viscous Parameter ($\delta$) | Turbulence Parameter ($\xi$) |
|---|---|---|---|---|
| Calibration Range | 1° to 45°[1], Step: 1° | 0.25 to 0.8, Step: 0.1 | 1e7 to 5e11[2] | 100 to 7600, Step: 100 |
| lgl Rutsch | 23° | - | 5e9 | - |
| Insel, 1 stage | 28° | 0.55 | - | 4600 |
| Insel, 2 stages | 28° | 0.48 | - | 4600 |

[1]For lgl Rutsch, a more restricted range of 20° to 30° was used.

[2]For the viscous parameter, the following values were used: (1e$x$, 2.5e$x$, 5e$x$) for $x$ = (7,8,9,10,11)

## 4 Results

### 4.1 lgl Rutsch

The best fit parameters for the lgl Rutsch event are summarized on Table 1. Overall, it was found that the available velocity data provides a strong constraint on the best fit viscous parameter, and the deposit distribution well constrains the friction angle. The best fit simulation results are shown on Figure 6, where it can be seen that the overall impact area (solid black) is



well reproduced by the simulation. Too much material is deposited on the proximal slope in source zone, as compared to the field estimate, and the distal deposits are generally thinner than that estimated from the pre-event topo reconstruction.

Timelapse deposit depths taken every 6 months (Figure 7) shows that the mass spreads quickly during the first year, before reaching a steady state velocity, and finally coming to rest after about three years. This is also shown on the simulated velocities (Figure 3), which compare well with those inferred from eye witness accounts in terms of the trend and the absolute values. Our simulations have initially high velocity values, on the order of 1 m/day, which well matches those reported by Heim (1881). They then slow down, again in accordance with the previously cited account, before movement reaches levels of a

few cm/day, and eventually ceasing altogether.

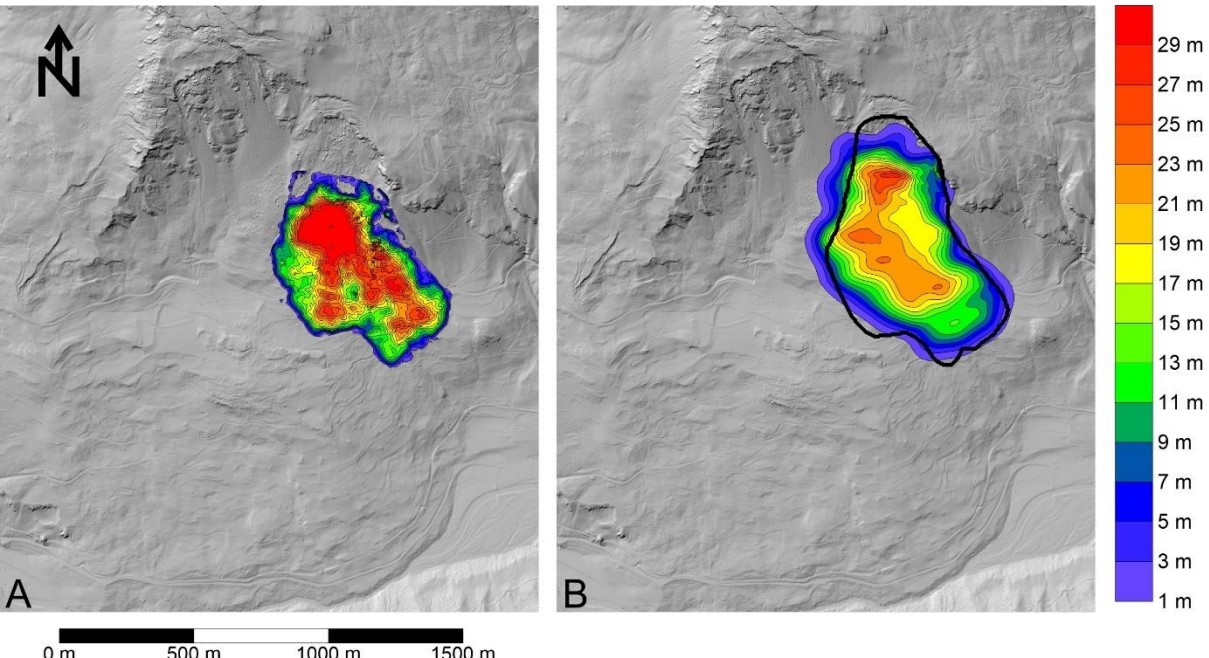

**Figure 6: A) Estimated deposit thicknesses based on field evidence. B) Best fit simulation results, with the black line showing the impact area estimated from the post-event hillshade. Elevation data used to generate hillshade: © swisstopo.**





**Figure 7: Timelapse simulation results, showing the emplacement of the lgl Rutsch landslide. The black outline shows the impact area constrained from field observations. Elevation data used to generate hillshade: © swisstopo.**



## 4.2 Insel

The best fit parameters for the different failure Insel scenarios are shown in Table 1, and the corresponding simulation results using the frictional and Voellmy rheologies are given on Figure 8. The simulated material distribution for the one and two
stage results is shown on Figure 9. All simulations can reproduce the observed deposit distribution reasonably well, albeit with differences in the volume of material deposited in the source zone. Furthermore, both drag laws lead to very similar results, because the best fit turbulence coefficient is high. Figure 9 shows that the simulation scenarios result in different distributions of the source zone lithologies, with the two-stage results simulating deposition of Allgäu schist in the source zone, and a mixing of Allgäu schist and Vallatscha dolomite at the toe, whereas the single-stage simulations predict mainly
Vallatscha dolomite in the source zone, and Allgäu schist at the toe. This difference occurs because, in the one stage simulations, the upslope Vallatscha compresses the downslope Allgäu units, leading to most of the Allgäu vacating the source zone. Geological interpretations of the post-failure situation indicate that the source zone deposit is mainly Vallatscha dolomite (Loew et al., 2024), and the test pits in the debris indicate that the toe is primarily composed of highly disturbed Allgäu schist (Figure 4). This suggests that the single-stage failure scenario better matches the observed distribution of lithologies in the
deposit.

Timelapse simulation results are shown on Figure 10 for the single-stage frictional simulations. As can be seen, the simulation takes about 50 seconds to come to rest, and leaves a significant deposit in the source zone, as observed based on the field evidence. It should be noted that the thick deposit along the margins of the source zone are an artefact of the interpolation methodology used in SPH.





**Figure 8: Final deposit distribution simulated for the various simulation scenarios. A) Deposit distribution estimated from the field evidence, B) Two stage frictional, C) One stage frictional, D) One stage Voellmy. The thick deposits on the margins of the source zone are an artefact of the interpolation algorithm used in the numerical model. Elevation data used to generate hillshade: © swisstopo.**







**Figure 9: Comparison of lithology distribution for the two-stage and one-stage simulations. Red: mainly Allgaü Schist and to a minor extent Raibler Rauwacke, Blue: Vallatscha Dolomite. The inset shows the pre-failure distribution of the two main lithologies at the source zone at t = 0. Elevation data used to generate hillshade: © swisstopo.**





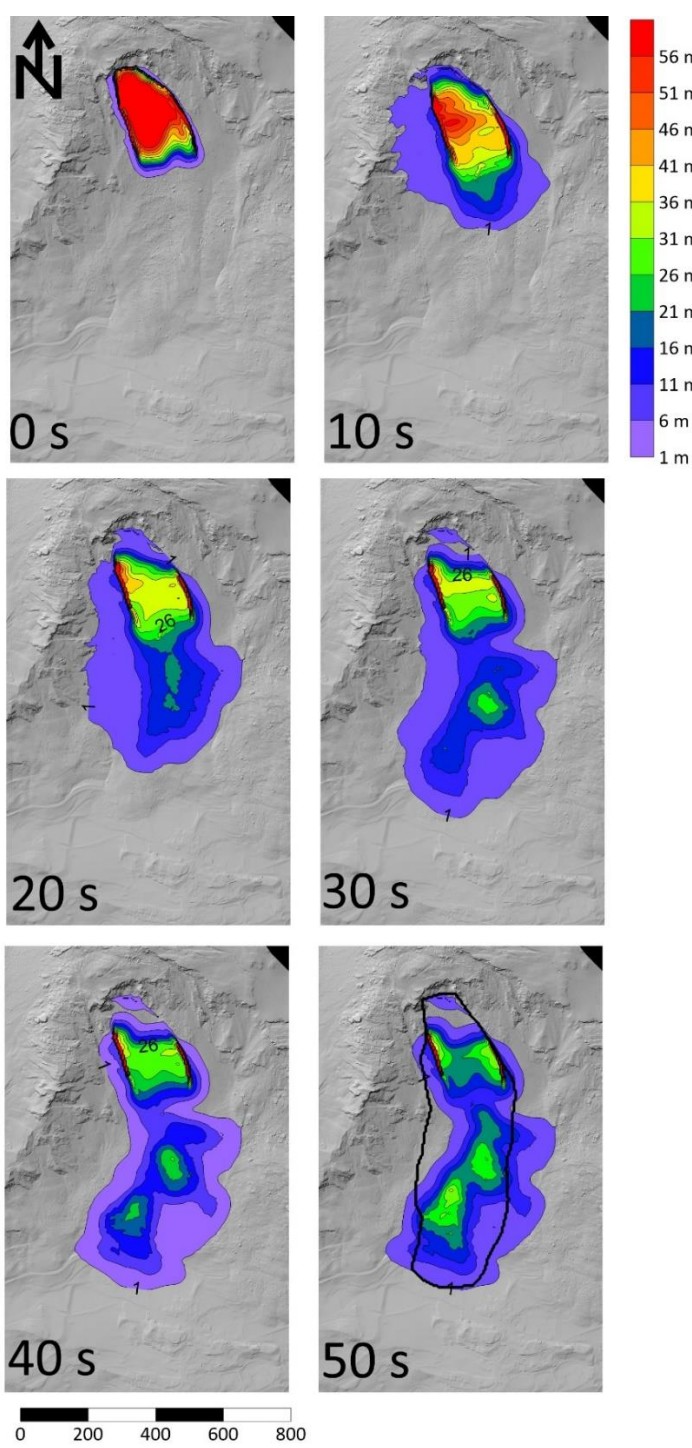

**Figure 10: Timelapse simulation of the best-fit results for the one-stage simulations. The black outline shows the impact area constrained from field observations. Elevation data used to generate hillshade: © swisstopo.**



## 5    Discussion

### 5.1    Semi-Implicit Numerical Model

Numerical models of landslide motion have typically been implemented to simulate the motion of extremely-rapid, flowlike
landslides (e.g. Aaron & Hungr, 2016; Hungr, 1995; Hungr & McDougall, 2009). Few researchers have attempted to simulate
the motion of slower events, and those that have often use simplified 1D models that cannot account for spreading (e.g. Ranalli
et al., 2010). This is likely because of the extremely small timesteps required by explicit solvers that are needed to avoid
unphysical oscillations in the simulated velocities that lead to numerical instabilities. These timesteps would result in decadal
to century simulation times using a CPU implementation of our numerical model.

We have overcome these numerical constraints by using a highly optimized GPU implementation of SPH, and a semi-implicit
timestepping scheme. Our validation indicates that the assumptions we make in the derivation are appropriate for simulating
the rheology we use to attain the moderate to rapid velocities. It is interesting to note that our simulations require the use of
both implicit timestepping and GPU computing. This is because our implicit timesteps cannot get so big, due to the
assumptions regarding the internal pressure distribution used to derive the governing equations. However, the use of a GPU
allows for relatively small timesteps (~720 s) to be used to solve three years emplacement duration in a few tens of minutes
using a desktop GPU (Nvidia Geforce RTX 3070 Ti).

The ability of our new numerical scheme to overcome the computational challenges described above is likely due to a few
unique characteristics that govern the emplacement of flowlike landslides. In particular, most source geometries feature a
much larger spatial area than height, leading to relatively small internal pressure gradients ($\frac{\partial h}{\partial x}$ and $\frac{\partial h}{\partial y}$ terms in Eq. [1] and [2]).
acceptable to use relatively large timesteps in the newly implemented implicit timestep scheme. With the model validated and
verified (Figure 6 and Figure 3), there is potential to simulate other flowlike landslides of this type, in particular earthflows
(e.g. Aaron et al., 2021; Keefer & Jonhnson, 1983; Mackey & Roering, 2011). This will be a subject of future work, and the
present implementation fills an important existing gap in the analysis of landslide motion.

### 5.2    lgl Rutsch

Large landslides in rock have been typically documented to experience extremely-rapid velocities (e.g. Aaron & McDougall,
2019; Coe et al., 2016; Hungr & Evans, 2004) or very slow to slow velocities (e.g. Ranalli et al., 2010; Wolter et al., 2020).
However, our qualitative and quantitative analyses of the lgl Rutsch landslide clearly show that large displacement, flowlike
landslides can occur in rock with velocities intermediate between these two end members. The mechanisms by which this can
occur are this critical for understanding the behaviour of other, similar landslides.

Our lgl Rutsch simulation results, which are verified by the eye witness accounts, suggest that this landslide experienced
significant viscous resistance during its emplacement, which lead to its low velocities and three-year emplacement duration





(Figure 3). We infer that the source of the viscous resistance experienced by this landslide is the clay-rich Allgäu Schists which are present in the source zone (Figure 2). Our back-analysis results further imply that these units can experience relatively low dynamic friction angles (~23°).

The simulated mass comes to rest due to a combination of spreading (which reduces internal pressure gradients), and changes in the topographic slope angle. Remarkably, it was not necessary to account for pore pressure generation and dissipation to explain the emplacement of the lgl Rutsch failure, suggesting that it behaved differently from classical earthflows, despite superficial similarities (e.g. Mackey & Roering, 2011). However, our simulations do not reproduce seasonal variations in the landslide behaviour, which are described in the eye-witness accounts and which are still present in the recent deformation
behavior measured in some of the boreholes. This likely could be addressed by employing time-varying parameters in our proposed basal drag law (Eq. [5]), though this would complicate the calibration procedure.

Our simulations result in too much deposition in the source zone, as compared with that inferred from the topographic reconstruction (Figure 6). This could be caused by uncertainties in the topographic reconstruction, as well as the dam break initial condition that is used in our model, whereby it is assumed that the mass is instantaneously in a state of distributed shear
failure. In reality, the mass likely moved some distance as a rigid body, with internal failure and flowlike movement only occurring sometime later, along with loosening and softening of the failed material which transformed it from more intact schist into flowing soil that is rich in clay (e.g. Coe et al., 2016). Additionally, the eastern part of the moving mass is transported over stable ground, whereas the western part is transported over the moving mass of the Rutschung Dorf, which may have reworked the deposit after the event.

**5.3    Insel**

Our analysis clearly demonstrates that, compared to many other large volume catastrophic rock slope failures which transitioned into rock avalanches (e.g. Aaron, 2017; Aaron & McDougall, 2019), the Insel event exhibited low mobility. This is demonstrated by its high H/L value (0.6), the observed limited runout beyond the toe of the slope (Figure 10), and best-fit Voellmy resistance parameters that are much more resistive than typical for rock avalanches (Aaron & McDougall, 2019).
Previously, the relative low mobility of some events had been explained with reference to disintegration into a series of small volume failures over a long time (e.g. Eberhardt et al., 2004). However, our analysis of available monitoring data, as well as our simulation results, support an interpretation of the Insel event whereby the majority of the mass failed during an ~2 minute period starting at approximately 23:37 (local time). Our results further suggest that, if multiple failures happened, they were not separated substantially in time and featured significant dynamic interaction, which is necessary to reproduce the observed
deposit distribution. This interpretation is consistent with the seismic data, as it shows a pronounced spike at this time, the georadar, which shows downslope decorrelation at this time, as well as the simulation results, which show a better fit to observed deposit lithologies if the event is simulated as a single slope failure. We therefore interpret the thin mantling of



blocks of Vallatscha dolomite on the deposit (Figure 4) as the result of dynamic interaction and overriding of the upslope Vallatscha unit onto the downslope Allgäu unit.

Our interpretation of these results is that they provide a clear demonstration of the importance of extrinsic factors, specifically the path conditions, in governing flowlike landslide mobility. Aaron et al. (2022) recommends dividing the potential runout path into a 'source' and 'path', and assessing the shear strength differently in these two zones. In the case of Insel, it appears as though no sudden weakening occurred in the source zone, which is consistent with the large pre-failure displacements. Further, the path material encountered was not composed of weak substrate, such as saturated, liquefiable material, snow or
ice, which would have significantly enhanced the mobility of the event. This suggests that, if the failure had happened when the site was snow covered, or when the colluvium on site was saturated, the dynamics could have been different. The Insel failure thus serves as an important end member case history when assessing rock slope mobility in the context of a hazard analysis, as it demonstrates that large volume catastrophic rockslope failures do not necessarily exhibit excessive mobility.

### 5.4    Event Comparison and Implications for Runout Analysis

Our detailed analysis of the lgl Rutsch and Insel failures has demonstrated remarkable differences between the two events, despite their source zones being located a mere 500 m apart, within the same geologic sequences and on the same slope. Whereas lgl Rutsch took three years to emplace, Insel took only 2 minutes, a difference of five orders of magnitude. The likely difference between these two failures is the proportion of clay-rich Schist present within the failed mass. At lgl Rutsch, the majority of the mass is in these clay-rich units (Figure 2), which likely resulted in its viscous shear behaviour. Conversely,
Insel contained a large portion of brittle dolomites from the Vallatscha formation (Loew et al., 2024, p. 20), and consequently underwent a more catastrophic failure process as compared to the lgl Rutsch. Interestingly, the Insel rupture surface daylighted in clay-rich Schist, which allowed it to accommodate the observed large pre-failure displacements (> 50 m) that occurred prior to failure (Kenner et al., 2025, p. 202). These results therefore highlight the critical importance of accounting for source zone lithology when understanding and forecasting the future behaviour of deforming rock slopes and their emplacement after
failure. They further highlight the sensitivity to relatively minor changes in the proportion of ductile and brittle units in the source zone.

### 6    Conclusion

We have presented the derivation, implementation and validation of a new numerical model for simulating the motion of moderate to rapid flowlike landslides. We applied this model to the historic lgl Rutsch landslide, which took three years to
emplace. This was contrasted with the (observed and simulated) behaviour of the 2023 Insel landslide, which emplaced in two minutes. The main findings of our work are summarized below:



1. The assumptions made in the derivation of the semi-implicit numerical scheme appear valid for landslides which move with velocities on the order of meters per day. The numerical approach used herein has wider applicability to many types of landslide movement as 3 years of movement time can be simulated in a few tens of minutes.

2. Our back-analysis of the lgl Rutsch events suggests that this landslide experienced significant viscous resistance during its emplacement. This is likely caused by the high proportion of clay-rich schist units present in the source mass.

3. In contrast, the Insel event featured a smaller proportion of clay-rich units, and moved with extremely-rapid velocities. Analysis of the available field data, as well as our numerical modelling results, supports an

interpretation whereby the mass failed as one main stage over an approximately 2 minute time period.

4. Our analysis clearly demonstrates that not all large volume, catastrophic rockslope failures experience excessive mobility. The lack of excess mobility exhibited by Insel is likely due to the large amount of pre-failure deformation combined with the high-strength, dry substrate that the mass overrode. The relatively low mobility cannot be explained by failure sequence.

5. The difference in behaviour between the lgl Rutsch and Insel landslides demonstrates an acute sensitivity of landslide velocity to source lithologies. Despite relatively moderate changes in the proportion of clay-rich rocks in the source zone, these two landslides displayed velocities that varied by five orders of magnitude.

These results have substantial implications for hazard management of rockslopes in many geological settings, where engineers and practitioners are tasked with forecasting runout given that a catastrophic failure has occurred. Typically, scenarios are

defined where a failure is either a catastrophic rock avalanche, or a progressive disintegration of the mass in a series of rockfall and dry granular flows. The failures documented at this site represent two landslide classes rarely considered, but clearly indicate scenarios which must be considered when forecasting landslide hazard at similar sites.

## 7 Acknowledgements

We are grateful to Christian Nagy and Andreas Huwiler from the AWN Graubunden for interesting conversations and

substantial research support. We also thank Reto Grischott and Flurina Brunold of BTG AG for their support and interesting conversations regarding this work. This work benefited from the freely available data provided by the Swiss Seismological





Service. This work was funded in part by a research contact given to the Chair of Engineering Geology from AWN Graubunden. JL acknowledges funding support from the NERC (NE/X00029X/1).

## 8    Data Availability

The seismic data used can be found at the following reference:

- Swiss Seismological Service (SED) At ETH Zurich; (2012): Temporary deployments in Switzerland associated with landslides; ETH Zurich. Other. https://doi.org/10.12686/SED/NETWORKS/XP

- BRIZ1: https://doi.org/10.3929/ethz-b-000677205

- BRIZ2: https://doi.org/10.3929/ethz-b-000671357

Upon acceptance, the data used in this study will be available from the ETH research data collection.



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
