# Peer review of "Dynamic Analysis of Flowlike Landslides at Brienz/Brinzauls, Graubünden, Switzerland"

_EGUsphere, 2025_

## Author Response (AR1)

**RC2**: 'Comment on egusphere-2025-2788', Anonymous Referee #2, 31 Jul 2025

This paper presents a detailed dynamic analysis of two flow-like landslides (Igl Rutsch and Insel) in Brienz/Brinzauls, Switzerland. It introduces a novel GPU-accelerated numerical model (Orin-3D) capable of simulating landslide velocities ranging from moderate (meters/day) to extremely rapid (>5 m/s). The research is innovative, methodologically sound, and well-supported by field data, historical records, and modern monitoring techniques. The study provides significant theoretical and practical contributions to landslide hazard assessment and risk prediction. The paper is well-structured, logically organized, and supported by appropriate figures and references. Overall, the manuscript is of high academic merit and merits publication. Several suggestions are listed below for reference.

Thank you so much for your kind words and great comments.

1. The paper notes that pore pressure dynamics were not considered in the simulations, which could be significant for clay-rich landslides. A more detailed discussion of this assumption's implications should be added.

Thanks for this important comment, we have updated the discussion to read (new text in bold):

*The simulated mass comes to rest due to a combination of spreading (which reduces internal pressure gradients), and changes in the topographic slope angle. Remarkably, it was not necessary to account for pore pressure generation and dissipation to explain the emplacement of the lgl Rutsch failure.* **Most landslides with moderate to rapid velocities, such as earthflows (e.g. Mackey & Roering, 2011), are acutely sensitive to pore pressure dynamics. The fact that we could reproduce many features of lgl Rutsch without explicit consideration of pore pressure suggests its dynamics are different from these events, despite superficial similarities. However, Figure 3 shows that, between Jan 1879 and July 1881, the landslide likely went through seasonal acceleration phases, which are still present in the inclinometer data measured in some of the boreholes (albeit at much smaller magnitudes). This suggests that seasonal accelerations due to pore pressure dynamics may overprint the bulk behaviour, and that this is not captured in our numerical simulations.** *This could potentially be addressed by employing time-varying parameters in our proposed basal drag law (Eq. [5]), though this would complicate the calibration procedure.*

2. The model does not fully capture the seasonal velocity variations observed in the Igl Rutsch landslide. The authors should explore incorporating time-dependent parameters (e.g., viscous coefficient δ) to better match field observations.

We agree with this comment, and have more clearly highlighted and contextualized this limitation in the updated text above. However, we consider incorporating time dependent parameters beyond the scope of the present work, due to a lack of pore pressure data that

could be used to justify the selected values.  We therefore prefer to keep our simulations as simple as possible, so that we can interpret the results to infer dynamics.

We updated the paragraph below in section 5.1 to address this important point:

*The ability of our new numerical scheme to overcome the computational challenges described above is likely due to a few unique characteristics that govern the emplacement of flowlike landslides.  In particular, most source geometries feature a much larger spatial area than height, leading to relatively small internal pressure gradients ($\frac{\partial h}{\partial x}$ and $\frac{\partial h}{\partial y}$ terms in Eq. [1] and [2]), making it acceptable to use relatively large timesteps in the newly implemented implicit timestep scheme.  With the model validated and verified (**Error! Reference source not found.** and **Error! Reference source not found.**), there is potential to simulate other flowlike landslides of this type, in particular earthflows (e.g. Aaron et al., 2021; Keefer & Jonhnson, 1983; Mackey & Roering, 2011).  **In particular, pore pressure data is occasionally available for these other landslide types, which could justify the use of more complex, time dependent rheological parameters.**  This will be a subject of future work, and the present implementation fills an important existing gap in the analysis of landslide motion.*

3. A sensitivity analysis of key parameters (e.g., δ and φ) would strengthen the model's interpretability.

This is a great point, and we summarize the results of a sensitivity analysis using the methodology described in Aaron et al. (2019) on Figure 1 below.  As can be seen, the friction angle governs the impact area, and restricts the values of φ to ~ 23° to 26°.  The viscous parameter mainly governs velocity, and provides the best fit for δ ~= 5e9.  We decided not to include this Figure in the manuscript because we think it would overcomplicate the methods and results, and that these are already described in Section 5.2.

[Figure]

*Figure 1: Sensitivity analysis of the key rheological parameters. Left: parameter posterior probability when only the trimline is used. Middle: parameter posterior probability when only the velocity data is used. Right: combined posterior probability density function.*

**RC3**: 'Comment on egusphere-2025-2788', Anonymous Referee #3, 13 Aug 2025

This study addresses an important and specialized type of landslide—large failures in clay-rich rocks that transition into flow-like movements. The authors present a depth-averaged SPH model with a novel rheology capable of simulating slow-moving landslides (velocities on the order of m/day) and conduct detailed back-analyses of two well-documented events at Brienz/Brinzauls, which exhibited emplacement velocities differing by five orders of magnitude. The topic is highly relevant, and the manuscript is well-structured and clearly written. I recommend minor revisions before publication in NHESS.

Thanks for taking the time to provide this feedback, we really appreciate it.

Specific Comments:

1. The Lgl Rutsch event was simulated using 4,000 SPH particles. Could the authors clarify the rationale for this choice and discuss whether the results are sensitive to particle resolution?

This is a fair point, which we should have justified better. 4,000 particles was selected to trade-off simulation resolution and model runtime, which we wanted to keep low as we use a calibration methodology that requires hundreds to thousands of model runs. We updated section 3.1 to read:

*We back-analysed the lgl Rutsch and Insel case histories in order to derive model parameters that can be used for forecasting events that have similar types. For lgl Rutsch, we used the new semi-implicit runout model (Equations **Error! Reference source not found.**) and **Error! Reference source not found.**), as well as the failure geometry shown on **Error! Reference source not found.**. We used timesteps of 720 seconds, 4000 particles, a topographic resolution of 10 m, and performed a probabilistic calibration (described below) in order to*

*calibrate the friction angle and viscous parameter (Equation (5)).* **We used a numerical resolution of 4,000 particles as the results are relatively insensitive to this choice (Aaron, 2023), and it keeps model runtimes low.** *As part of this calibration, we compared measured and simulated velocities for the event (**Error! Reference source not found.**). The latter velocities were obtained by calculating the velocity of the simulated front as it moves over the slope, as this provides the closest simulated quantity to the velocities estimated from eye witness accounts.*

2. The adopted SPH model is depth-averaged, which simplifies the real 3D dynamics. Please discuss how this assumption might influence the model's predictive accuracy, particularly for flow transitions and velocity distributions.

Thanks for this comment, we have updated Section 3 to read (new text in bold):

*In the present work we use the numerical model 'Orin-3D' as the computational framework with which to analyse the lgl Rutsch and Insel failures. Orin-3D is a GPU accelerated, depth-averaged Lagrangian model that solves the equations of motion using a parallel implementation of the smooth particle hydrodynamics (SPH) numerical method (Aaron, 2023).* **The use of a depth-averaged model is justified in the present cases due to the fact that both case histories have a much greater planar extent compared to their thickness, and the topography is relatively regular. However, we note that our model cannot resolve the vertical velocity distribution of the flow, and instead computes a depth-averaged velocity.** *Orin-3D provides an over two orders of magnitude increase in computational efficiency, compared to a widely used equivalent fluid model (Aaron, 2023).*

3. Table 1 reports best-fit friction angles of 23°–28° for different stages. However, field and experimental studies (e.g., rock avalanches, earthflows) typically suggest much higher friction coefficients. Note that the reviewer is mainly working on fast-moving geophysical flows (e.g., Kong et al., JGR, 2023, 10.1029/2022JF006870). Are these smaller values physically justified for clay-rich rocks?

This is a great point. For the upper range (28°), this value is close to that expected for the dynamic friction coefficient of flowing fragments of rock (Heim; Hsu). For the clay rich rocks, our value is within the range determined for other studies *(**Ranalli et al., 2010; Skempton et al., 1997**)* We updated the text as follows:

In section 5.2:

*Our lgl Rutsch simulation results, which are verified by the eye witness accounts, suggest that this landslide experienced significant viscous resistance during its emplacement, which lead to its low velocities and three-year emplacement duration (**Error! Reference source not found.**). We infer that the source of the viscous resistance experienced by this landslide is the clay-rich Allgäu Schists which are present in the source zone (**Error! Reference source not found.**). Our back-analysis results further imply that these units can experience*

*relatively low dynamic friction angles (~23°),* **although these are consistent with some other reported values (Ranalli et al., 2010; Skempton et al., 1997).**

In section 5.3:

*Our analysis clearly demonstrates that, compared to many other large volume catastrophic rock slope failures which transitioned into rock avalanches (e.g. Aaron, 2017; Aaron & McDougall, 2019), the Insel event exhibited low mobility,* **consistent with that expected for a dry granular flow of rock fragments (e.g Heim, 1932).** *This is demonstrated by its high H/L value (0.6), the observed limited runout beyond the toe of the slope (***Error! Reference source not found.***), and best-fit Voellmy resistance parameters that are much more resistive than typical for rock avalanches (Aaron & McDougall, 2019). Previously, the relative low mobility of some events had been explained with reference to disintegration into a series of small volume failures over a long time (e.g. Eberhardt et al., 2004). However, our analysis of available monitoring data, as well as our simulation results, support an interpretation of the Insel event whereby the majority of the mass failed during an ~2 minute period starting at approximately 23:37 (local time). Our results further suggest that, if multiple failures happened, they were not separated substantially in time and featured significant dynamic interaction, which is necessary to reproduce the observed deposit distribution. This interpretation is consistent with the seismic data, as it shows a pronounced spike at this time, the georadar, which shows downslope decorrelation at this time, as well as the simulation results, which show a better fit to observed deposit lithologies if the event is simulated as a single slope failure. We therefore interpret the thin mantling of blocks of Vallatscha dolomite on the deposit (***Error! Reference source not found.***) as the result of dynamic interaction and overriding of the upslope Vallatscha unit onto the downslope Allgäu unit.*

4. For the Insel event (Section 4.2), were the same SPH parameters (fitted to Lgl Rutsch) applied? If so, this seems inconsistent given the five-order magnitude velocity difference between the two events. Please clarify whether material or rheological differences explain this discrepancy.

For this event we use a different, non-viscous rheology, with different rheological parameters. This results in the substantial difference in the simulated velocities of the two events, which are consistent with observations.

**RC1**: 'Comment on egusphere-2025-2788', Anonymous Referee #1, 22 Jul 2025

The authors present a numerical model to simulate flow-like landslides from unstable rock slopes over a broad range of velocities (five orders of magnitude), and apply their model for

the back-calculation of two events in the landslide area of Brienz, Switzerland. This topic is highly relevant from both a scientific and a practical point of view, and perfectly suits to the scope of the journal. The preprint is generally well written, structured, and illustrated, and I would certainly like to see this work published. I suggest some minor revisions based on the comments below, which might help to further improve an already very good paper.

Thanks for your constructive and thorough review.

- There are some minor issues of language and formatting (e.g., in L176 and L197, the closing brackets after the equation numbers are missing).

Thanks for catching this, we fixed these examples and checked the rest of the text for similar typos.

- Section 3.1: past tense and present tense are mixed, there should be consistent use of one of them.

Great catch, we have fixed this.

- Caption of Fig. 9: Allgaü Schist -> Allgäu Schist

Fixed

- 7 and Fig. 10: in Fig. 7, the observed outline is included in all panes, in Fig. 10 only for the last time step – maybe better to make this consistent.

We agree, and have changed the figure as shown below:

[Figure]

- L290: You should also include work by others than Hungr and colleagues.

This is a great point which we completely agree with.  We have updated this line to read:

Numerical models of landslide motion have typically been implemented to simulate the motion of extremely-rapid, flowlike landslides **(e.g. Aaron & Hungr, 2016; Bouchut et al., 2003; Christen et al., 2010; Hungr, 1995; Hungr & McDougall, 2009; Pirulli, 2005; Pudasaini & Mergili, 2019)**

- L290f: There have been attempts of slow-flow simulation models also in GIS environments: the r.avaflow software now has a function for slower movements: https://link.springer.com/article/10.1007/s10346-023-02146-z and https://egusphere.copernicus.org/preprints/2025/egusphere-2025-213/ Recently, a dynamic earth  flow model was published: https://www.sciencedirect.com/science/article/pii/S0013795225000559

Thanks for pointing us to this important reference, we will add it to the introduction and discussion.

- L396: I am not sure about the statement "The failures documented at this site represent two landslide classes rarely considered …". I think landslides of such type are often considered by practitioners. Maybe better reformulate the statement to make clearer what you mean.

We agree, and upon reading your comment have realised that this statement was poorly formulated.  We have rephrased as below (new text in bold):

***These results thus show that consideration of source zone lithology and mobility enhancement factors, such as the presence of saturated substrate, must be incorporated in the analysis of landslide hazard at similar sites.  As the two analysed cases show, ignoring these factors could lead to substantial overestimates of both impact area and velocity.***

- L402: research contact --> research contract

Fixed, thanks!

---

## Author Response (AR2)

Thanks very much for your thorough edits!  We really appreciate it.  I've made all the changes to the bibliography requested, and have a few notes below.  Regarding the bibliography, we changed the referencing style to use the template provided by Copernicus, to hopefully better conform with the Journal's standards.

- In Fig. 2, there are some schematics for the borehole KB12/21, respectively, the blue and cyan vertical lines and horizontal signs. What do they represent? Should you put them in the legend?

This is a great catch, thanks!  We have updated the Figure as below.  These different lines represent the water tables in the stable bedrock and Igl Rutsch landslide body, respectively.

[Figure]

- Please check the sentences where formula are given as see if : are needed;

We checked and corrected this.  In particular we deleted a colon before the reference to Eqn 6, and added one before Eqn 7.

- Could you check, for example, line 390 from the revised manuscript, whether the "IGL" is written correctly or if it is with a small L instead of I?

We checked and corrected all references in the manuscript.  Your right that it should be capital I and not small l.

- reference Bouchut, F., Mangeney-Castelnau, A., Perthame, B., & Vilotte, J.-P. (2003). A new model of Saint Venant and 455 Savage–Hutter type for gravity driven shallow water flows. Comptes Rendus Mathematique, 336(6), Article 6. https://doi.org/10.1016/S1631-073X(03)00117-1 has the doi link broken and pages 531-536

I double checked this but I think the link is correct.  The link leads to an institutional website, but I checked this on the journal homepage and the DOI provided there also leads to the same institutional page.  I therefore assume that the link is correct.

- reference Casulli, V. (1990). Semi-implicit finite difference methods for the two-dimensional shallow water equations. Journal of Computational Physics, 86(1), 56–74. https://doi.org/10.1016/0021-9991(90)90091-E has the doi link broken

As above, I double checked it but I think the link works.